# Progressing Ultragreen, Energy-Efficient Biobased Depolymerization of Poly(ethylene terephthalate) via Microwave-Assisted Green Deep Eutectic Solvent and Enzymatic Treatment

**DOI:** 10.3390/polym14010109

**Published:** 2021-12-29

**Authors:** Olivia A. Attallah, Muhammad Azeem, Efstratios Nikolaivits, Evangelos Topakas, Margaret Brennan Fournet

**Affiliations:** 1Materials Research Institute, Technological University of the Shannon: Midlands Midwest, N37 HD68 Athlone, Ireland; oadly@ait.ie (O.A.A.); mfournet@ait.ie (M.B.F.); 2Pharmaceutical Chemistry Department, Faculty of Pharmacy, Heliopolis University, Cairo-Belbeis Desert Road, El Salam, Cairo 11777, Egypt; 3Biotechnology Laboratory, Industrial Biotechnology & Biocatalysis Group, School of Chemical Engineering, National Technical University of Athens, 15780 Athens, Greece; stratosnikolai@gmail.com (E.N.); vtopakas@chemeng.ntua.gr (E.T.)

**Keywords:** enzymatic hydrolysis, deep eutectic solvents, polyethylene terephthalate, Box-Behnken design, microwave depolymerization

## Abstract

Effective interfacing of energy-efficient and biobased technologies presents an all-green route to achieving continuous circular production, utilization, and reproduction of plastics. Here, we show combined ultragreen chemical and biocatalytic depolymerization of polyethylene terephthalate (PET) using deep eutectic solvent (DES)-based low-energy microwave (MW) treatment followed by enzymatic hydrolysis. DESs are emerging as attractive sustainable catalysts due to their low toxicity, biodegradability, and unique biological compatibility. A green DES with triplet composition of choline chloride, glycerol, and urea was selected for PET depolymerization under MW irradiation without the use of additional depolymerization agents. Treatment conditions were studied using Box-Behnken design (BBD) with respect to MW irradiation time, MW power, and volume of DES. Under the optimized conditions of 20 mL DES volume, 260 W MW power, and 3 min MW time, a significant increase in the carbonyl index and PET percentage weight loss was observed. The combined MW-assisted DES depolymerization and enzymatic hydrolysis of the treated PET residue using LCC variant ICCG resulted in a total monomer conversion of ≈16% (*w/w*) in the form of terephthalic acid, mono-(2-hydroxyethyl) terephthalate, and bis-(2-hydroxyethyl) terephthalate. Such high monomer conversion in comparison to enzymatically hydrolyzed virgin PET (1.56% (*w/w*)) could be attributed to the recognized depolymerization effect of the selected DES MW treatment process. Hence, MW-assisted DES technology proved itself as an efficient process for boosting the biodepolymerization of PET in an ultrafast and eco-friendly manner.

## 1. Introduction

All-green routes to continuous circular material and commodity production, unmaking and remaking in a manner analogous to nature’s many resource cycles, remain largely elusive for plastics [1]. Polyethylene terephthalate (PET) plastic value chain is a pertinent example of many current linear mine, use, and dispose economic processes. PET is highly recalcitrant and widely used in the manufacturing of packaging materials, beverage bottles, and synthetic fibers due to its high mechanical and thermal properties, nontoxicity, and excellent transparency [2]. The unabated increase in the demand for PET production is a grave environmental concern given the poor degradation rates of PET in soil and air [3]. Mechanical and chemical processing are the current mainstay approaches for PET recycling, with each having considerable limitations [4]. Loss of transparency of mechanically recycled PET and the presence of traces of reactive antimony catalyst restrict the application of recycled PET in food and beverage packaging [4,5]. On the other hand, chemical recycling, which comprises glycolysis, methanolysis, aminolysis, and hydrolysis to depolymerize PET into its monomers [6,7,8,9,10], requires long reaction times, large volumes of non-green solvents for reaction, and several product purification processes [11]. Recently, a number of alternative techniques are being explored for PET depolymerization, including the incorporation of efficient catalytic systems in depolymerization reactions [12,13], supercritical technology [14], and microwave-assisted methods [15,16]. However, despite the achievement of increased reaction rates, the need for harsh reaction conditions and use of non-green solvents remain a considerable challenge [17,18]. Recently, plastic biodepolymerization has been proposed as an environmentally friendly and promising technology for PET recycling [19]. Greener approaches for PET recycling, such as complete solubilization of PET in natural deep eutectic solvents and thin-layer film synthesis from PET polymer waste for nanofiltration, have also been employed recently as sustainable routes for PET recycling [18,20,21]. Herein, a novel multistep approach that echoes nature’s sequential depolymerization steps for naturally occurring polymers, namely weathering, arthropodal digestion, and microbial and enzymatic degradation, is presented. A MW-assisted DES technique was combined with enzymatic hydrolysis to obtain enhanced PET depolymerization compared with enzymatic hydrolysis alone. Such a recycling methodology would be advantageous due to its low energy requirements and operation under mild conditions during plastic degradation/depolymerization [22]. A series of impeding factors, namely the need for low physical dimension preparations of the polymer as suitable substrates, slow catalytic activity, enzymatic thermal degradation at high processing temperatures [20], low interaction levels with the chemical structures of linear polymers [23], and high polymer crystallinity and hydrophobicity [24], serve to hinder the efficiency of biobased plastic recycling. Assisting techniques designed to overcome these barriers, which can render the plastic more amenable to biodepolymerization/biodegradation and augment the probability of depolymerization events using biobased agents, are required to progress towards sustainable plastic resource cycling. Recently, new combinations of physiochemical treatment techniques have been applied to overcome existing hindrances to biodepolymerization [25]. For instance, Falah et al. [26] proposed several sequential physiochemical treatments, including ultraviolet, high temperature, and nitric acid solvent treatment, prior to exposing PET for enzymatic degradation. The authors observed the development of cracks on the PET surface after treatment, which led to some enhancement in the enzymatic degradation of PET. Quartinello et al. [27] used a sequential chemoenzymatic treatment to facilitate depolymerization of PET from textile waste under mild conditions. The chemical treatment was performed under neutral conditions (pressure = 40 bar and temperature = 250 °C) to depolymerize PET into high-purity terephthalic acid (TPA) and small oligomers with a total monomer conversion of 85% within 90 min. Enzymatic hydrolysis was then performed using Humicola insolens cutinase to yield 97% pure TPA. Furthermore, Gong et al. [28] used a combination of alkaline hydrolysis and microbial strains (T = 37 °C, pH = 12, time = 48 h) and found enhanced conversion of PET into its functional monomers as a result of faster microbial growth and reduction in particle size of PET. Deep eutectic solvents (DESs) are a new class of ionic liquids that are becoming prominent for plastics depolymerization due their unique characteristics [1]. The use of DESs as catalysts in depolymerization reactions can make reaction conditions milder and decrease reaction times [29]. Recently, these solvents have been utilized as catalysts in microwave (MW)-assisted PET depolymerization reactions due to their strong MW heating characteristics and the synergic hydrogen bond formation of these solvents with PET polymer chains [8,15]. Different compositions of DESs have been employed and evaluated for the enhancement of PET depolymerization under mild conditions, as elaborated in Appendix A. To the best of our knowledge, the combination of MW-assisted DES technique and enzymatic hydrolysis to obtain enhanced PET depolymerization compared to enzymatic hydrolysis alone has not been previously explored and presents a strong progress towards the achievement of all-green permanent resource circularity in tandem with nature. Noticeably, treating PET in DESs under MW irradiation can increase PET chain flexibility and change the physicochemical properties of the polymer [6]. Thus, such MW-assisted DES treatment can provide an enhanced monomer conversion yield upon PET depolymerization. 

In this study, an all-green, environmentally friendly sequential PET depolymerization approach was employed comprising treatment of PET using MW-assisted DES technique without the use of additional depolymerization agents followed by enzymatic hydrolysis using a variant of LCC cutinase. A green DES of ternary composition of choline chloride, glycerol, and urea was selected for PET treatment in the presence of MW irradiation. Recently, the Box-Behnken design (BBD) has been utilized in many studies for the optimization of reaction processes [30,31]. In this work, optimized MW treatment conditions were determined using BBD with respect to MW irradiation time, MW power, and volume of DES. The crystallinity index, carbonyl index, and weight loss of residual PET were used as the studied responses for BBD. Residual PET resulting from the optimized MW treatment process was further exposed to a four-day hydrolysis process by a thermostable polyesterase, and the total depolymerization efficiency was evaluated. The success of the combined techniques proposed in this study is expected to provide a green, environmentally friendly approach for plastic recycling.

## 2. Materials and Methods

### 2.1. Materials

PET granules were purchased from Alpek Polyester UK Ltd. (Lazenby, UK) and converted into micron-sized fine powder using a centrifugal miller (Retsch Verder Scientific, Haan, Germany). Glycerol (99%), choline chloride (98%, ChCl), and urea (98%) were purchased from Sigma-Aldrich (Dorset, UK). All other chemicals were obtained from Aldrich (Darmstadt, Germany) and were of analytical grade and readily available to use without any purification.

### 2.2. Preparation of DES

The synthesis of the ternary DES was based on the method provided by [32]. Prior to the preparation of DES, ChCl was dried overnight at 65 °C in the oven. A DES of triplet composition based on urea/glycerol/ChCl was synthesized with 1:1:1 molar ratio by continuously mixing and heating at 80 °C until a homogeneous, clear, and transparent liquid was formed within 10 min.

### 2.3. MW Treatment Experiments

The MW treatment experiments were carried out by mixing 1 g of powdered PET in varied volumes of synthesized DES while stirring for 15 min. The prepared suspensions were then exposed to MW irradiation at specified MW power and time. After MW treatments, residual PET was filtered and washed three times with distilled water to obtain clean residual PET, and DES was regenerated. The PET residues were then dried in an oven at 70 °C overnight and kept in sealed containers for further analysis. 

### 2.4. Experimental Design 

A three-factor, three-level Box-Behnken design (BBD) (Design Expert Stat-Ease Inc., Minneapolis, MN, USA) was implemented for optimization of the proposed MW-assisted DES technique. A total of 15 runs with three center points were set up to study the following three factors: MW irradiation time (min) (X_1_), microwave power (W) (X_2_), and volume of DES (ml) (X_3_). The responses were concluded as the weight loss (%), carbonyl index, and crystallinity index of residual PET. The chosen values for the studied factors were constructed using reported literature and preliminary experiments (Table 1). 

### 2.5. Enzymatic Hydrolysis of PET Materials

For the enzymatic depolymerization, LCC variant ICCG (LCCv) was used [33]. The coding sequence of LCCv was codon optimized for expression in *E. coli* and cloned into pET26b(+) vector (GenScript Biotech B.V., Leiden, the Netherlands). The expression and purification of the recombinant protein was performed as described previously [34]. The purity of the resulting enzymatic preparation was checked on SDS-PAGE electrophoresis (12.5% (*w/v*)), and protein concentration was determined by measuring the absorbance at 280 nm based on the calculated molar extinction coefficient.

MW-treated PET samples used for enzymatic depolymerization were washed twice with ultrapure water in order to remove residual monomers that could potentially inhibit enzymatic action. Reactions took place in 10 mL of 100 mM potassium phosphate buffer, pH 8, containing 100 mg of PET residue and 4 μΜ of enzyme. Control reactions without the addition of enzyme were also realized. All reactions were incubated at 55 °C under shaking for 4 days. After that, 0.1% (*v/v*) of 6M HCl was added in each reaction and centrifuged at 4000× *g* at 10 °C. Supernatants were collected and analyzed by HPLC (Perkin Elmer, Boston, MA, USA) [34] in order to determine the concentration of the resulting water-soluble degradation products. The remaining material was washed 3 times with ultrapure water, freeze-dried, and weighed. Experiments were run in triplicates, and the standard deviation was estimated.

### 2.6. Instrumental Characterization

The PET samples before and after MW-assisted DES treatment were analyzed by FTIR spectroscopy (Perkin Elmer, Washington, MA, USA) at a spectral region of 4000–600 cm^−1^. The carbonyl index was determined based on the obtained results using the baseline method. Ratios of ester carbonyl peak intensity at 1713 cm^−1^ to that of the normal C–H bonding mode at 1408 cm^−1^ in PET were calculated as follows [35]:(1)$$Carbonyl\;index=\frac{Absorption\;at\;1713{\;\mathrm{cm}}^{-1}}{Absorption\;at\;1408{\;\mathrm{cm}}^{-1}}$$

The thermal behavior of the samples was evaluated by a DSC Perkin Elmer 4000 (Perkin Elmer Washington, MA, USA) with Pyris Software version 13.3.1 (Perkin Elmer Washington, MA, USA) under an inert nitrogen stream. About 10 mg of specimen was sealed in an aluminum pan. The DSC scans were recorded while heating from 30 to 275 °C at a heating rate of 10 °C min^−1^ and then cooled to 30 °C. The crystallinity index was calculated according to the following equation [36]:Crystallinity index = (∆H_m_/W∆H_m0_) × 100(2)
where ∆H_m_ (Jg^−1^) is the heat of fusion of the PET sample, ∆H_m0_ is the heat of fusion for completely crystalline PET (140 Jg^−1^) [37], and W(g) is the weight fraction of residual PET in the samples.

The percentage weight loss of PET was determined at onset temperature of degradation (T_0_) using a thermogravimetric analyzer Pyris TGA (Perkin Elmer, Washington, MA, USA). The polymer samples were placed in a standard aluminum pan and heated from 30 to 600 °C at the rate of 10 °C min^−1^ under nitrogen flow of 50 mL min^−1^. The PET weight loss (%) was calculated as follows:PET weight loss (%) = (100 − weight percent of PET at T_0_)(3)

## 3. Results

### 3.1. Properties of DES 

Ternary DES has been reported as a new type of DES to widen the range of DES applications owing to the extra functionality provided by their components. One pertinent area where ternary DESs were recently applied is CO_2_ capture, which demonstrates that DESs have great flexibility in terms of synthesis, forms, and applications [32]. In the current study, we selected a green ternary DES with ChCl as a hydrogen bond acceptor and glycerol and urea as hydrogen bond donors to provide an initial depolymerization of PET when coupled with MW irradiation and to facilitate PET enzymatic hydrolysis. A schematic diagram of the proposed interactions of DES with PET is demonstrated in Figure 1. The advantage of this DES lies in its components being green, inexpensive, and largely available with the capacity to provide synergistic effects on PET depolymerization [7,15,27,33]. As shown in Figure 1, depolymerization of PET is postulated to involve a form of glycolysis reaction due to the presence of glycerol within the ternary DES [15]. It is also known that a quaternary ammonium compound, such as ChCl and DES itself, could act as a catalyst in mild glycolysis [7,8,15,26]. Simultaneously, the H-bond action between glycerol and urea is expected to change the charge density of the hydroxyl (OH) group in glycerol and increase the electronegativity of the oxygen atom in the glycerol OH group. Hence, the nucleophilicity of the oxygen becomes stronger, thereby supporting preferential attack to the carbon of the ester group in PET [38,39]. 

Based on previous reports, DESs with high pH values and low viscosity can contribute to the catalytic activity and influence the reaction rate [39,40]. The pH and density values of the proposed ternary DES were found to be 9.95 and 1.16 g/mL, respectively, giving rise to a suitable pH for depolymerization reaction to occur. Moreover, the employment of MW irradiation in the current depolymerization technique led to the production of expeditious heating, in particular with DES due to its high electric conductivity [41]. Therefore, initial depolymerization of PET can be achieved very efficiently due to the synergistic effects of MW irradiation, glycolysis due to glycerol, and the catalytic activities of ChCl, urea alone, and in DES. 

Prior to the depolymerization process, the prepared DES was also characterized using FTIR, as shown in Figure 2. In the typical FTIR spectrum of pure urea, the characteristic C=O, N–H and C–N stretching peaks appeared at 1677, 3427.12, and 1459.79 cm^−1^, respectively, while the N–H deformation peak appeared at 1590.21 cm^−1^. Pure glycerol’s spectrum showed a C–O stretching peak at 1043.55 and 1111.08 cm^−1^ and O–H stretching peak at 3339.48 cm^−1^. The FTIR spectrum of pure ChCl had O–H and C–H stretching peaks at 3220.59 and 3006.97 cm^−1^, respectively, and the asymmetric and symmetric stretching peaks of C–N linkage were observed at 954.54 and 892.96 cm^−1^, respectively. The FTIR spectrum of the prepared DES showed the formation of new bonds at 1668.38 and 1624.66 cm^−1^ compared to the FTIR spectra of pure urea, glycerol, and ChCl. The C=O linkage frequency peak at 1677 cm^−1^ of urea was shifted to lower side at 1668 cm^−1^, indicating the formation of more hydrogen bonds, and the N–H stretching frequency of urea at 3427.12 cm^−1^ was masked by the O–H stretching peak [42]. The C–O stretching peak of glycerol at 1043.55 cm^−1^ was also shifted to higher wavenumber in the DES, indicating the successful formation of DES.

### 3.2. Experimental Design Results

The model of PET MW treatment was studied using response surface methodology. In the current study, the experimental runs were carried out based on the design plan proposed for the studied parameters (MW irradiation time, MW power, and volume of DES). After each run, the crystallinity index, carbonyl index, and weight loss (%) at T_0_ of degradation of treated PET were calculated. The results are presented as responses for each run in Table 2.

The studied responses were then tested against different regression models to determine the best-fitting mathematical model and the significance of varying the process parameters. The quadratic model was chosen as the best fitting model for the studied responses in comparison to the other models. The relationship between the crystallinity index (Y_1_), carbonyl index (Y_2_), and weight loss of PET at T_0_ of degradation (Y_3_) and the studied parameters of MW irradiation time (X_1_), MW power (X_2_), and volume of DES (X_3_) is demonstrated in Table 3.

For the crystallinity index (Y_1_), the coefficients of the quadratic model equation indicated that the increase in all the studied factors led to a significant increase in the crystallinity index of residual PET except for the volume of DES, where the *p*-value was more than 0.05. The interaction between MW irradiation time and volume of DES showed a positive effect on the crystallinity index as well revealed the significant effect both factors had on the PET residues. Moreover, the MW power interactions with both the MW irradiation time and volume of DES also showed a significant positive efficacy on the crystallinity index. Such results indicate that all the studied factors and their interactions had positive effects on the crystallinity index of treated PET, with the increase in the studied factors leading to an increase in the degradation of PET and causing an increase in the crystallinity index of the treated samples [43]. 

For the carbonyl index (Y_2_), as demonstrated in Table 2, all treated PET residues had higher carbonyl index than that of the untreated PET (2.80). Both MW irradiation time and power showed a significant negative effect on the carbonyl index values, while the volume of DES showed a positive effect. The MW irradiation time interactions with both MW power and volume of DES also showed significant negative effects on the carbonyl index of PET residue. Thus, based on the obtained results and the carbonyl index of untreated PET, the increase in both MW irradiation time and power led to a degree of PET depolymerization, which was observed through the low values of the carbonyl index of the residual PET. On the other hand, high levels of DES volume with low levels of MW irradiation time and power caused a significant increase in the carbonyl groups on the surface of the PET as a result of surface oxidation rather than complete depolymerization of treated PET.

For PET weight loss at T_0_ of degradation (Y_3_), as elaborated in Table 3, the coefficients of the model equation showed that MW irradiation time and power and their interaction with each other had positive effect on PET weight loss. On the other hand, the volume of DES and its interactions with both MW irradiation time and power showed negative efficacy on PET weight loss at T_0_ of degradation. These results indicate that increased levels of MW irradiation time and power and low levels of DES induce a decrease in thermal stability of treated PET, leading to a greater degree of PET depolymerization at the T_0_ of degradation. 

The adequacy of the proposed model to describe the crystallinity index, carbonyl index, and weight loss of treated PET at T_0_ of degradation was evaluated, and the results are demonstrated in Table 3. Based on the statistics test, high coefficients of determination were observed for all responses. The adjusted R^2^ values were calculated to be 0.9944 for the crystallinity index, 0.9966 for the carbonyl index, and 0.9941 for the percentage weight loss of PET at T_0_ of degradation.

Analysis of variance (ANOVA) was also applied to determine the significance of the model at a 95% confidence interval. A model is said to be significant if the probability value (*p*-value) is <0.05. The *p*-values demonstrated in Table 3 indicate that all the studied responses fitted the model well. From the lack-of-fit test. the response showed a highly desirable nonsignificant lack-of-fit (*p* > 0.1) with *p*-values of 0.228 for the crystallinity index, 0.355 for the carbonyl index, and 0.537 for the percentage weight loss of PET at T_0_ of degradation.

### 3.3. Response Surface Analysis

Response surface graphical plots were generated between the responses obtained for PET MW treatment and the studied independent variables to estimate the effect of combinations of these variables on the studied responses. The 3D and contour plots for the crystallinity index, carbonyl index, and weight loss of PET at T_0_ of degradation are demonstrated in Figure 3, Figure 4 and Figure 5, respectively. 

As shown in Figure 3, high levels of both MW irradiation time and power caused an increase in the crystallinity index of PET residue. Such result can be attributed to the initial degradation of PET upon treatment with the MW-assisted DES technique, which usually occurs in the amorphous phase of PET, causing an increase in the overall crystallinity of the polymer [43]. A significant increase in the crystallinity index was also observed with the increase in DES volume until 35 mL. Further increase in DES volume did not show a profound effect on the crystallinity index, indicating that low volumes of DES are more effective in the initial depolymerization of PET using the proposed MW treatment technique. 

The carbonyl index is considered one of the critical responses used in the evaluation of the hydrophilic nature of treated polymers. Hydrophilic polymers with high values of carbonyl index are more amenable for microbial depolymerization than hydrophobic ones. Figure 4 illustrates the dependence of the carbonyl index on the studied factors of MW irradiation time, MW power, and volume of DES. It can be observed that the increase in both MW irradiation time and power did not cause a significant increase in the carbonyl index of treated PET samples. Thus, it can be assumed that high levels of MW irradiation time and power result in PET depolymerization rather than surface oxidation, which is in accordance with the crystallinity index results. On the other hand, the interactions of high levels of DES volume with low levels of both MW irradiation time and power showed a significant increase in the carbonyl index, reaching a value of 4.55 and confirming the process of PET surface oxidation at these levels of the studied factors. 

PET weight loss at T_0_ of degradation is also an important response for assessing the initial depolymerization of PET. The higher the value of weight loss, the greater was the decrease in the thermal stability of the treated polymer, which confirmed the initial depolymerization of the treated PET samples. In Table 3, it can be observed that all the independent variables and their interactions influenced PET weight loss significantly (term *p*-value < 0.05) except for the volume of DES. As indicated in Figure 5, the interaction between MW irradiation time and MW power resulted in a significant increase in PET weight loss at T_0_ of degradation to reach a value of 9%, whereas the value of weight loss of untreated PET was measured to be only 0.44%. In addition, it should be noted that low levels of DES volume showed higher PET weight loss percentage than high levels of DES volume, which means that initial depolymerization of PET occurs better at low volumes of DES. 

### 3.4. Optimization of the MW-Assisted DES Technique 

All three responses were optimized simultaneously using BBD optimization. Optimum MW treatment conditions were chosen with the aim of obtaining maximum initial depolymerization of PET and enhancing the biodegradation of residual PET after MW treatment. As previously described, maximum initial depolymerization of PET was observed with PET residues of increased percentage of weight loss at T_0_ of degradation. Additionally, based on literature review, enhanced PET biodegradation can be achieved through low crystallinity and high carbonyl index of residual PET [44]. Thus, the MW treatment conditions were adjusted to attain minimum crystallinity index and maximum carbonyl index and percentage weight loss of PET at T_0_, as shown in Table 4. Based on BBD, a total of 63 optimized solutions were obtained. The selected solution was determined according to its success in attaining an acceptable desirability of >0.5 for the studied responses and in fulfilling the low carbon footprint goal with the lowest energy consumption concerning MW irradiation time and power. A batch experiment was carried out for PET MW treatment using the optimized conditions while keeping PET concentration at 1.0 g, and the three responses were evaluated to validate the predicted model factors and responses. The response values (predicted and observed) for the optimized conditions are recorded in Table 4. The model was proven to be validated as a fine agreement existed between the predicted and observed results. This indicates the success of BBD for the evaluation and optimization of the proposed PET treatment process. 

### 3.5. Enzymatic Depolymerization of PET Materials

Combination of green treatments with enzymatic hydrolysis as means for enhanced plastic recycling has been very limited within the literature. In this work, a multistep depolymerization process for PET comprising a green treatment process followed by enzymatic hydrolysis using a highly PET-active enzyme was performed. The total depolymerization of PET obtained after the optimized MW treatment and enzymatic hydrolysis was compared against that of untreated virgin PET biodegradation. The polymer’s biodegradability was assessed based on the percentage weight loss of the material and the amount of the produced water-soluble monomers, namely TPA, mono-(2-hydroxyethyl) terephthalate (MHET), and bis-(2-hydroxyethyl) terephthalate (BHET), analyzed via HPLC (Appendix A). Following the MW treatment process, the optimized PET sample showed 20.2 ± 1.4% weight loss, and 15.76% (*w/w*) of the treated PET sample corresponded to a mixture of the released soluble monomers TPA, MHET, and BHET. Alternatively, the average percentage weight loss (%) for untreated and MW-treated PET samples was estimated to be 1.5 ± 0.3 and 1.8 ± 0.3%, respectively, after enzymatic hydrolysis. Such low percentage of weight loss can be attributed to the crystallinity of the materials where there are a few accessible amorphous regions for the enzyme to act on. Slightly higher weight loss percentage was observed for the MW-treated PET sample, which indicated the efficiency of MW treatment in enhancing the biodegradability of PET.

As detailed in Table 5, the total amount of monomers released after enzymatic hydrolysis was slightly higher for the virgin PET than for MW-treated PET residue (0.82 versus 0.55 mM). In both cases, 60–70% of the total products released was in the form of TPA and 30–35% as MHET, while only 1% remained as BHET. The higher product release for virgin PET could be explained by its slightly lower crystallinity index (31.40) compared to MW-treated PET residue (32.98), a factor that enhances enzyme action. 

Moreover, the proposed multistep depolymerization approach resulted in the production of monomers during both the MW treatment and enzymatic hydrolysis processes, giving rise to an average PET weight loss of 22 ± 1.7% and a total monomer conversion of ≈16% (*w/w*). Thus, the obtained results for the suggested depolymerization protocol were considerably higher than the virgin PET undergoing enzymatic hydrolysis only. 

## 4. Conclusions

A stepwise depolymerization process for PET comprising an all-green, fast, low-energy, MW-assisted DES technique without the use of additional depolymerizing agents followed by enzymatic hydrolysis using LCCv enzyme was demonstrated. Compared to virgin PET biodepolymerization, the demonstrated combined approach was able to achieve increased PET depolymerization with a total of ≈16% (*w/w*) monomer conversion. The developed MW treatment process was optimized using BBD, where the volume of DES, MW power, and MW irradiation time were studied as independent variables. FTIR, TGA, and DSC spectra of the residual PET obtained after treatment with the MW-assisted DES technique showed a significant increase in residual PET carbonyl index and percentage weight loss at T_0_ of degradation and maintenance of PET crystallinity percentage. Furthermore, optimum MW treatment was obtained at low DES volume (20 mL), 260 W MW power, and 3 min MW irradiation time. The enzymatic hydrolysis of treated PET demonstrated 1.8% weight loss and 0.55 mM monomers released after enzymatic hydrolysis, while 1.5% weight loss and 0.82 mM monomers were recorded for virgin PET. Analysis of the recycled monomers using HPLC confirmed the presence of TPA, MHET, and BHET as the monomers produced in the treated samples. The isolation of these monomers will be done in future work. The combination all-green treatments, which operated under mild, low-energy conditions without the use of additional depolymerization agents, produced an average PET weight loss of 22 ± 1.7% and a total monomer conversion of ≈16% (*w/w*). This MW-assisted DES followed by enzymatic hydrolysis methodology shows strong potential to achieve high conversion rates and is amenable to the incorporation of additional and sequential green approaches.

Moreover, large-scale applications of MW-assisted depolymerization in a continuous manner has recently shown great potential for recycling as it facilitates depolymerization of a large amount of materials in relatively mild conditions (lower temperature and frequencies) [45]. Nevertheless, there are certain challenges with respect to high-cost reactor designs, emission of volatile degradation products, unequal irradiations due to hot spots, and nonuniform heating that still need to be resolved [46].

In conclusion, the promise of ultramild routes with no requirement for additional depolymerization agent is demonstrated herein for their capacity to play an instrumental role in highly sustainable degradation and depolymerization processes for PET and other polyesters, thus serving as a key step in delivering ultrasustainable all-green routes for circular plastic value chains. 

## Figures and Tables

**Figure 1 polymers-14-00109-f001:**
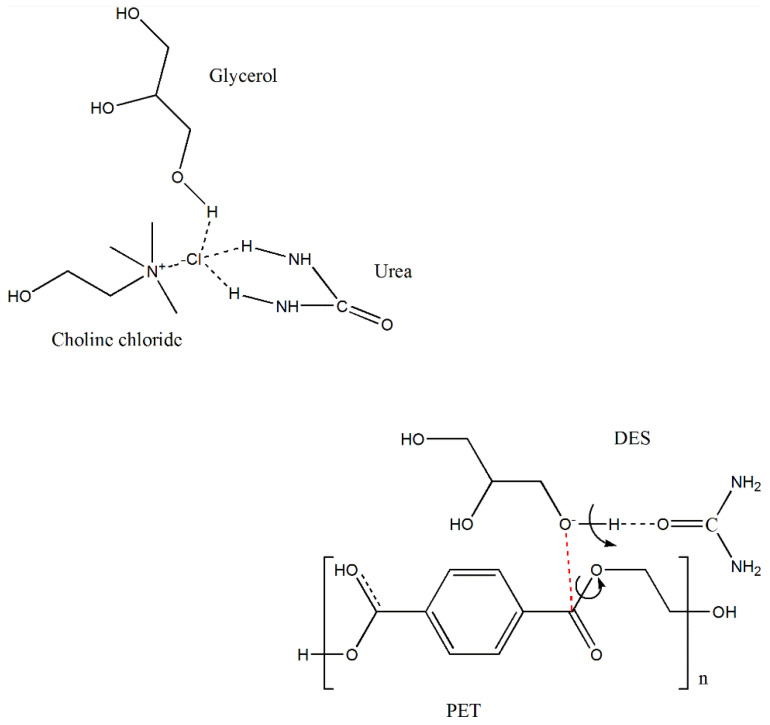
Schematic diagram for the proposed interactions of DES (composed of choline chloride/urea/glycerol in the ratio of 1:1:1) with PET via hydrogen bonding.

**Figure 2 polymers-14-00109-f002:**
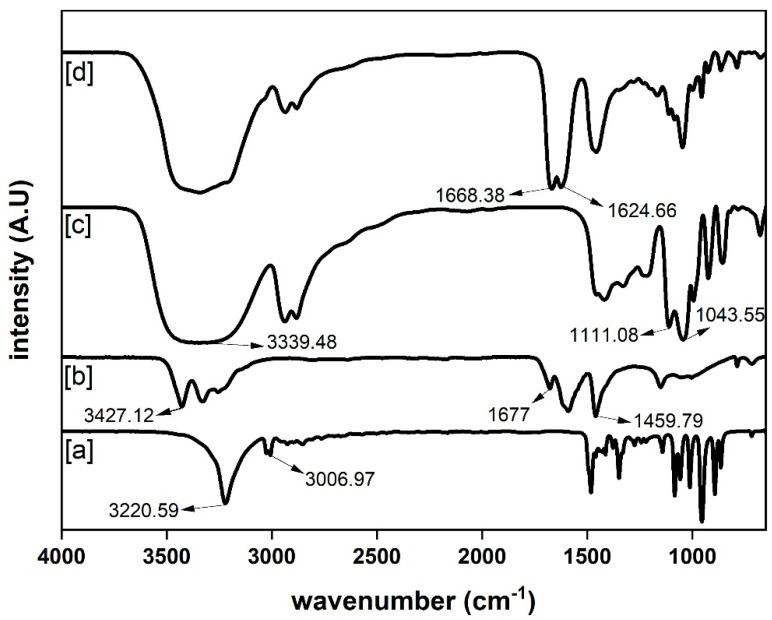
FTIR spectra of: (**a**) choline chloride, (**b**) urea, (**c**) glycerol, and (**d**) DES (choline chloride/urea/glycerol in the ratio of 1:1:1).

**Figure 3 polymers-14-00109-f003:**
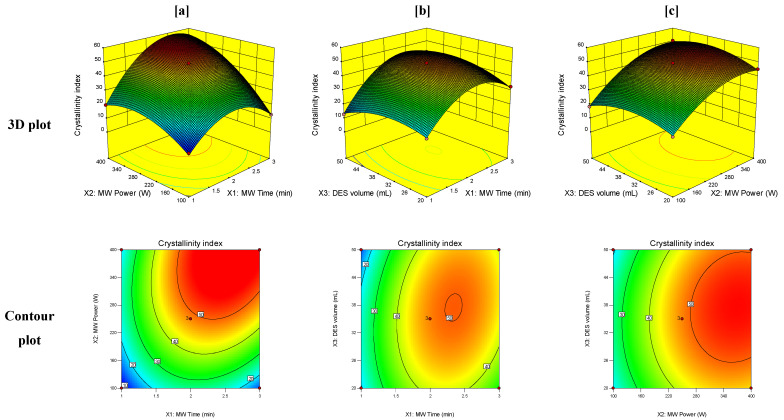
3D and contour plots of the effect of the interaction of (**a**) MW time (X_1_) and MW power (X_2_), (**b**) MW time (X_1_) and volume of DES (X_3_), and (**c**) MW power (X_2_) and volume of DES (X_3_) on the crystallinity index.

**Figure 4 polymers-14-00109-f004:**
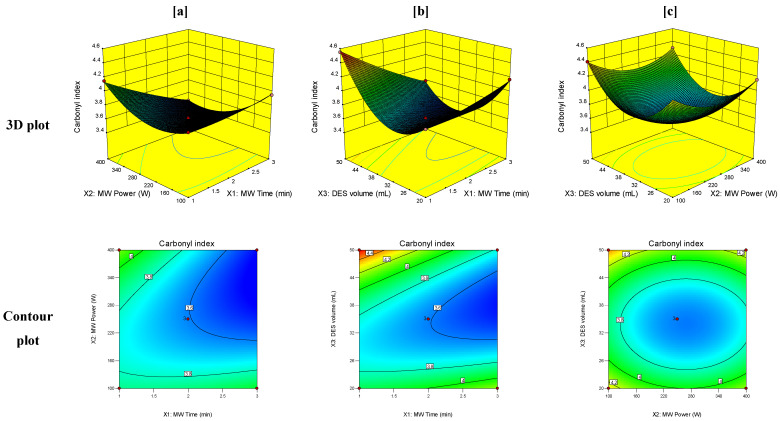
3D and contour plots of the effect of the interaction of (**a**) MW time (X_1_) and MW power (X_2_), (**b**) MW time (X_1_) and volume of DES (X_3_), and (**c**) MW power (X_2_) and volume of DES (X_3_) on the carbonyl index.

**Figure 5 polymers-14-00109-f005:**
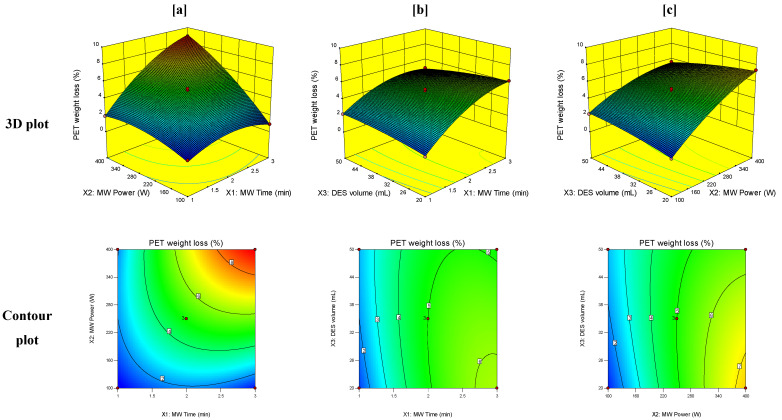
3D and contour plots of the effect of the interaction of (**a**) MW time (X_1_) and MW power (X_2_), (**b**) MW time (X_1_) and volume of DES (X_3_), and (**c**) MW power (X_2_) and volume of DES (X_3_) on PET weight loss (%).

**Table 1 polymers-14-00109-t001:** Variables and levels in Box-Behnken experimental design for PET pretreatment.

	Level			
Independent Variables	−1	0	1	Constrains
X_1_: MW time (min)	1	2	3	In the range
X_2_: MW power (W)	100	250	400	In the range
X_3_: Volume of DES (mL)	20	35	50	In the range

**Table 2 polymers-14-00109-t002:** Experimental matrix and observed responses for PET pretreatment in BBD.

	Independent Variable			Dependent Variable		
Run	X_1_ (min)	X_2_ (W)	X_3_ (mL)	Y_1_	Y_2_	Y_3_ (%)
1	3	400	35	54.10	3.47	8.90
2	2	100	50	18.57	4.41	2.17
3	2	250	35	48.30	3.61	4.80
4	1	250	20	19.00	3.91	1.20
5	3	250	50	41.67	3.79	4.91
6	2	400	20	45.07	4.16	7.39
7	2	250	35	47.80	3.62	5.00
8	2	400	50	50.29	4.30	5.63
9	3	100	35	12.83	3.95	0.90
10	3	250	20	32.79	4.17	6.20
11	2	100	20	20.21	4.28	0.94
12	1	400	35	19.86	4.15	1.89
13	1	250	50	12.86	4.55	2.20
14	2	250	35	49.32	3.59	5.20
15	1	100	35	9.52	3.87	0.75

X_1_: MW irradiation time, X_2_: MW power, X_3_: volume of DES, Y_1_: crystallinity index, Y_2_: carbonyl index, and Y_3_: weight loss (%) at T_0_ of degradation.

**Table 3 polymers-14-00109-t003:** Statistical analysis of measured responses for PET pretreatment.

Fitting Model	Factors	Coefficient	*p*-Value	ANOVA
PET crystallinity index (Y_1_)	Intercept	48.47		*F* = 276.92, *R*^2^ = 0.9944, Model *p*-value < 0.0001, *p*-value of lack of fit = 0.228
	X_1_	10.02	<0.0001	
	X_2_	13.52	<0.0001	
	X_3_	0.79	0.1290	
	X_1_X_2_	7.73	<0.0001	
	X_1_X_3_	3.76	0.0017	
	X_2_X_3_	1.71	0.0385	
	X_1_^2^	−15.68	<0.0001	
	X_2_^2^	−8.72	<0.0001	
	X_3_^2^	−6.22	0.0002	
PET carbonyl index (Y_2_)	Intercept	3.61		*F* = 461.34, *R*^2^ = 0.9966, Model *p*-value < 0.0001, *p*-value of lack of fit = 0.355
	X_1_	−0.14	<0.0001	
	X_2_	−0.054	<0.0001	
	X_3_	0.066	0.0005	
	X_1_X_2_	−0.19	0.0002	
	X_1_X_3_	−0.25	<0.0001	
	X_2_X_3_	2.5 × 10^−3^	<0.0001	
	X_1_^2^	0.035	0.8048	
	X_2_^2^	0.22	0.0165	
	X_3_^2^	0.46	<0.0001	
Weight loss of PET at T_0_ of degradation (Y_3_)	Intercept	5.00		*F* = 264.71, *R*^2^ = 0.9941, Model *p*-value < 0.0001, *p*-value of lack of fit = 0.537
	X_1_	1.86	<0.0001	
	X_2_	2.38	<0.0001	
	X_3_	−0.10	0.2060	
	X_1_X_2_	1.72	<0.0001	
	X_1_X_3_	−0.57	0.0023	
	X_2_X_3_	−0.75	0.0007	
	X_1_^2^	−1.15	0.0001	
	X_2_^2^	−0.74	0.0008	
	X_3_^2^	−0.23	0.0825	

X_1_: MW irradiation time, X_2_: MW power, X_3_: volume of DES, Y_1_: crystallinity index, Y_2_: carbonyl index, and Y_3_: weight loss (%) at T_0_ of degradation.

**Table 4 polymers-14-00109-t004:** The optimized PET pretreatment process with observed and predicted response values.

Independent Variable		Optimized Level	
X_1_: MW time (min)		3.0	
X_2_: MW power (W)		260	
X_3_: Volume of DES (mL)		20.0	
Over all desirability		0.59	
Dependent variables	Desirability	Expected	Observed
Y_1_: PET crystallinity index	Minimize	33.39	32.98
Y_2_: PET carbonyl index	Maximize	4.14	4.22
Y_3_: PET weight loss (%)	Maximize	6.47	6.25

**Table 5 polymers-14-00109-t005:** Monomer concertation after a four-day incubation of LCCv enzyme with the untreated and treated PET materials.

Material	TPA (μΜ)	MHET (μΜ)	BHET (μΜ)
Untreated: Control	0.55 ± 0.04	0.00 ± 0.00	0.00 ± 0.00
Untreated: LCCv	521.13 ± 23.22	287.04 ± 7.63	7.07 ± 0.36
Treated: Control	0.83 ± 0.04	0.20 ± 0.00	0.00 ± 0.00
Treated: LCCv	384.79 ± 4.91	158.83 ± 4.52	4.21 ± 0.06

## Data Availability

All data generated or analyzed during this study are included in the article (and its Appendix A).

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
