# Peer review of "Progressing Ultragreen, Energy-Efficient Biobased Depolymerization of Poly(ethylene terephthalate) via Microwave-Assisted Green Deep Eutectic Solvent and Enzymatic Treatment"

_polymers, 2021, doi:10.3390/polym14010109_

Round 1

Reviewer 1 Report

The authors reported bio-based depolymerization of polyethylene terephthalate (PET) using microwave (MW) assisted deep eutectic solvents (DES) technique followed by enzymatic hydrolysis for PET recycling.

In their paper, a green DES of ternary composition of choline chloride, glycerol and urea was used for PET treatment assisted by MW, and the optimized MW treatment condition (MW irradiation time, MW power and volume of DES) were obtained by Box-Behnken design (BBD).

I think that the reported results in this paper are useful for the development of PET recycling.

There are some comments as follows.

  1. The intended meaning of the experiment for figure 6 should be mentioned in some detail.
  2. Line 30

The original word of the abbreviation ICCG should be attached.

  1. Line 119

BDD       ?

  1. Line 250

[a] Urea, [b] Choline chloride → [a] Choline chloride, [b] Urea     ?

Reviewer 2 Report

  1. The title should be simplified and buzz words eliminated, e.g. delete “ultra all-green, energy efficient”.
  2. The performance of the proposed methodology should be directly compared with recent approaches reported in the literature. What is the novelty of the work and what has been achieved? It is unclear how the field has been advanced and what is new, despite the clear potential impact of the work.
  3. Figure 1 should have a reference for the reaction mechanism. A proper reaction scheme should be drawn with starting materials on the left, arrow, products on the right etc.
  4. The author should discuss the potential and challenges of applying the microwave assisted methodology in a continuous manner, which is an important step forward.
  5. What was the rational for the selection of the Box-Behnken design for studying the treatment conditions with respect to MW irradiation time? What other designs were considered and deemed less appropriate?
  6. PET processing from a green perspective is important indeed and recent diverse efforts on this should be briefly mentioned to set the context before narrowing down the scope in the Introduction (10.1039/D0GC03226C; 10.1039/D1GC02403E).
  7. The isolation of the products and the recovery of the pure solvent should be demonstrated in the manuscript to complete the work. Full chemical characterization needs to be provided for them, and purity quantified.
  8. Recent PET depolymerization works should be acknowledged (10.1039/D1GC02896K; 10.1039/D1GC00887K; 10.1039/D1GC00665G).
  9. Some examples of ‘design of experiments’ such as the Box-Behnken design for reaction process optimization should be given (10.1016/j.matpr.2020.04.458; 10.1021/acscatal.8b01706).
  10. Line 135: include the actual time needed to achieve a homogeneous solution.
  11. Table 1 should be moved to the supporting information.
  12. Both the quotient (“x/y”) and negative exponent (“x y-1”) formats are used in the manuscript for units. Either of them should be used consistently, preferably the negative exponent format, which is recommended by the IUPAC.
  13. Product purity and isolation, chromatograms and spectra are entirely missing from the manuscript.

Round 2

Reviewer 2 Report

The comments are addressed.
